behaviour, neuroscience

quantal response equilibrium, bounded rationality, Prisoner's Dilemma, sensorimotor interactions, reinforcement learning

**Author for correspondence:**
Cecilia Lindig-León
e-mail: cecilia.lindig-leon@uni-ulm.de

[†]These authors contributed equally to this study.

# Bounded rational response equilibria in human sensorimotor interactions

Cecilia Lindig-León[†], Gerrit Schmid[†] and Daniel A. Braun

Institute of Neural Information Processing, Faculty of Engineering, Computer Science and Psychology, Ulm University, Germany

The Nash equilibrium is one of the most central solution concepts to study strategic interactions between multiple players and has recently also been shown to capture sensorimotor interactions between players that are haptically coupled. While previous studies in behavioural economics have shown that systematic deviations from Nash equilibria in economic decision-making can be explained by the more general quantal response equilibria, such deviations have not been reported for the sensorimotor domain. Here we investigate haptically coupled dyads across three different sensorimotor games corresponding to the classic symmetric and asymmetric Prisoner's Dilemma, where the quantal response equilibrium predicts characteristic shifts across the three games, although the Nash equilibrium stays the same. We find that subjects exhibit the predicted deviations from the Nash solution. Furthermore, we show that taking into account subjects' priors for the games, we arrive at a more accurate description of bounded rational response equilibria that can be regarded as a quantal response equilibrium with non-uniform prior. Our results suggest that bounded rational response equilibria provide a general tool to explain sensorimotor interactions that include the Nash equilibrium as a special case in the absence of information processing limitations.

## 1. Introduction

Sensorimotor interactions in humans include cooperative examples like carrying a table together across the room or dancing as well as competitive examples like arm wrestling, tug-of-war or playing tennis. There are many different frameworks to study sensorimotor interactions, including the sport sciences [1–3], the psychological sciences [4–18], the neurosciences [19,20] and even engineering when it comes to replicating successful sensorimotor interactions in human–machine interactions [21–24]. Quantitative concepts to study strategic interactions often originate from the decision sciences and include game theory [25–28] and reinforcement learning models [29,30] that were mainly developed in economics and computer science, respectively, but have also found application in studying sensorimotor interactions [24,31–35]. Without a doubt the central solution concept in the decision sciences to understand stable interaction patterns between different agents is the concept of the Nash equilibrium [36]. Roughly, a Nash equilibrium corresponds to a combination of strategies where no agent has anything to gain by deviating unilaterally from their strategy. Abstractly, a strategy can be conceived as a probability distribution over actions, so that Nash equilibria are in general pairs of distributions (mixed equilibria), or in special cases pairs of actions (pure equilibria), when the distributions concentrate their probability mass on a single action.

Previously, it was shown that sensorimotor interactions are amenable to the decision-theoretic framework of Nash equilibria [31]. By designing continuous sensorimotor versions of classic 2 × 2 matrix games like the infamous Prisoner's Dilemma, where players are haptically coupled and experience sensorimotor forces as pay-offs, it could be shown in several studies that human subjects naturally converge to Nash equilibria without verbal descriptions of the game [31–34]. While classic games are typically studied in behavioural

economics in cognitive decision-making tasks with explicitly communicated monetary pay-offs as utilities and clearly defined and known uncertainties, sensorimotor tasks typically involve implicit, action-related utilities such as motor effort or task accuracy and experiential probabilities that have to be learnt from many repetitions. Moreover, motor tasks often involve implicit learning, in contrast to explicit learning. When comparing the results of the sensorimotor games to the corresponding cognitive games, interesting differences can arise, as for example in the Prisoner's Dilemma where sensorimotor interactions regularly converge to the predicted Nash solution of defecting, whereas cognitive versions of the Prisoners' Dilemma regularly lead to some level of cooperation. Other studies have also found interesting differences between economic decision tasks and their equivalent sensorimotor tasks [37]. In particular, it has repeatedly been suggested that human sensorimotor behaviour abides by rational decision-making models [38], whereas for economic studies deviations from rational behaviour have been reported more frequently—although this idea has also been contested [39], and therefore requires further investigation.

While the Nash equilibrium is one of the most successful concepts in the decision sciences, it is also a well-known fact, in particular in behavioural economics, that human behaviour does not always perfectly align with predicted Nash equilibria [40,41]. It is safe to assume that there are multiple reasons for this failure depending on the exact tasks that are investigated, but one prominent reason that has been repeatedly proposed and quantitatively investigated in economic decision-making tasks is bounded rationality [42]. Players that are bounded rational are lacking perfect rationality required to maximize expected utility [43,44] in that they may not know all possible outcomes or the utility functions of the other players, they may have incomplete knowledge, model uncertainty or lack computational resources. One way to model limited information processing capabilities is to assume an information bound on how much players can change an *a priori* agnostic strategy (e.g. a uniform distribution over actions) to an expected utility maximizing strategy [45–48]. In the game-theoretic literature such information bounds on players' strategies have been investigated in the context of quantal response equilibria [49,50], which correspond to Nash equilibria in the unbounded limit, but can otherwise deviate significantly from Nash equilibrium solutions.

In behavioural economics, several studies have confirmed deviations from Nash equilibria in economic decision-making tasks that could be explained by quantal response equilibria [51–53], but so far it is unknown whether similar deviations can be observed in sensorimotor interactions. To this end, we designed three continuous sensorimotor versions of the traditional two-player matrix game of the Prisoner's Dilemma [54], corresponding to the classic symmetric form and two asymmetric variations. Crucially, all three versions of the game have the same single pure Nash equilibrium, but have different quantal response equilibria. In the classic symmetric form of the Prisoner's Dilemma it is assumed that both players can decide to either cooperate or to defect, but that regardless of what the other player decides, defecting is always associated with a better pay-off. The dilemma arises when both players follow this reasoning and end up with a pay-off that is worse compared to a situation where both players cooperate, but cooperation is unstable because each

player can improve their pay-off unilaterally by defecting. Intuitively, in the asymmetric version of the Prisoner's Dilemma, we can imagine that one of the prisoners has some form of weak alibi [55,56], that means that one player has more or less to lose than the other player when deviating from the stable Nash solution. Assuming that players have limited precision when maximizing expected utility due to bounded rationality, the quantal response equilibria in the asymmetric games differ systematically with the extra cost, even though the Nash equilibrium remains the same. This allows for a simple hypothesis test: does behaviour change across the three games or does it stay the same?

## 2. Results

In our sensorimotor version of the Prisoner's Dilemma, both players were sitting next to each other and used the handles of a robotic interface that each player could move freely in the horizontal plane—see figure 1*a* and electronic supplementary material, methods. During each trial, players were instructed to move their handle to touch the target bar that was projected onto a mirror above the plane of movement. The lateral position of both handles specified the individual magnitude of a resistive force exerted by the robot handles to oppose players' forward motion. Thus, we could induce a haptic coupling where the movement of both players affected the force as a form of pay-off experienced individually by each player in a continuous fashion. By imposing the three Prisoner's Dilemma pay-off landscapes shown in figure 1*b*, player 2 was always exposed to the same force landscape, whereas player 1 experienced more or less costs for deviating from the Nash solution depending on the condition. Accordingly, the quantal response equilibrium predicts a shift in player 1's equilibrium strategy, but not in player 2's, even though the Nash equilibrium is the same for all three games. To allow for learning of the different force landscapes, a particular haptic coupling was kept for a set of 40 trials. For our analysis in the following, we therefore focus on trajectory endpoints over such sets of 40 trials, where players can co-adapt.

Figure 2*a* shows a scatter plot of players' endpoint positions of the last 30 trials of all trial blocks in all three Prisoner's Dilemma games in relation to the Nash equilibrium at the position (1, 1) in the top right corner. The observed spread in the scatter plots suggests that behaviour differs across the three games. In particular, in the asymmetric high cost condition, deviations from the Nash equilibrium seem to be less than in the asymmetric low cost condition, where endpoints spread more freely in the entire plane. This impression from the scatter plots is confirmed when looking at the two-dimensional histograms in figure 2*b* that show a steeper increase in response frequencies for the asymmetric high cost condition and a shallower increase for the asymmetric low cost condition. This shift in the distribution across conditions can be quantified when determining the mean response of player 1 and player 2 from figure 2*a* and comparing it across conditions—see electronic supplementary material, figure S3. In particular, we find that the shift in player 1's strategy across conditions is highly significant ($p < 0.001$, rm ANOVA, $F = 24.86$, $d_1 = 2$, $d_2 = 14$), while player 2's strategy remains the same ($p > 0.1$, rm ANOVA, $F = 0.84$, $d_1 = 2$, $d_2 = 14$).

To analyse the results in terms of the classic Prisoner's Dilemma responses, we categorized the endpoints in figure 2*a*

**Figure 1.** Experimental set-up. (*a*) Sensorimotor game. Players were haptically coupled by two handles that generated a force on their hand resisting their forward movement. Importantly, each player's force depended on both players' lateral positions. (*b*) Pay-off matrices for each one of the three games including pay-offs for player 1 and player 2. The matrix values establish the boundary values of the force pay-offs at the limits of the $x_1 x_2$-space (left). Interpolation of the pay-offs defines a continuous pay-off landscape for a continuum of actions (right). (Online version in colour.)

according to their quadrants into Nash responses (defect–defect), cooperative responses (cooperate–cooperate) and exploitative responses (defect–cooperate or cooperate–defect). As can be seen in figure 2*c*, the Nash solution is predominant in all cases, although the exploitative response (cooperate–defect) becomes increasingly common across conditions as player 1's costs for deviating from the Nash equilibrium decreases. This change in strategy can also be seen in individual subject pairs in electronic supplementary material, figure S1, where the predominance of the Nash equilibrium response decreases across conditions.

To quantify this strategy shift in the probability space, we can determine how each player's response frequency $\lambda$, to choose the defect action, changes across conditions. As illustrated in figure 2*d*–*f*, player 1's shift in response frequency $\lambda$ is highly significant ($p < 0.001$, rm ANOVA, $F = 21.67$, $d_1 = 2$, $d_2 = 14$), while player 2's strategy remains the same ($p > 0.1$, rm ANOVA, $F = 0.48$, $d_1 = 2$, $d_2 = 14$). The response shift can also be observed at the level of individual pairs, although with higher variability—see electronic supplementary material, figure S2. In summary, we can conclude that the observed behavioural change across the three games is incompatible with the Nash equilibrium prediction of no change.

To see whether the observed deviations from the Nash equilibrium are consistent with the quantal response equilibrium hypothesis assuming limited information processing capabilities, we can compare the marginal distributions over responses of each player to the equilibrium distribution equations (2) and (3) in the electronic supplementary material predicted by the quantal response equilibrium when adjusting the precision parameters to fit the behaviour of subjects. Figure 3*a* shows the frequencies over players' responses in all games independent of the other player, separate for player 1 and player 2. As the cost structure for player 2 does not change, the quantal response equilibrium predicts that there should be no changes in the response distribution of player 2, which is in accordance with our data. For player 1, the quantal

response equilibrium predicts that in the asymmetric high cost condition, response frequencies should be elevated near the Nash equilibrium, but decreased further away from the Nash equilibrium compared to the classic symmetric Prisoner's Dilemma. Similarly, in the asymmetric low cost condition, the response frequencies around the Nash equilibrium should be suppressed, and instead elevated further away from the Nash solution. This prediction is confirmed, too, by the trends in the data. The only observation that is not predicted by the quantal response equilibrium is a border effect for player 1, where the response frequency in the direct neighbourhood of the Nash equilibrium declines sharply. However, such a border effect can be taken into account in the quantal response equilibrium model when considering non-uniform priors for the response distribution. To fit the players' empirical priors, we obtained a histogram over initial positions in the first trials of each block of 40 trials—figure 3*b*. It can be seen that player 1 assigns lower probabilities to the corners of the workspace and concentrates probability mass in the centre, whereas player 2 has a more uniform prior distribution. Taking into account these priors, the bounded rational equilibrium fits the data significantly better—compare figure 3*c*. The predicted mean of these fitted equilibrium distributions is also in good agreement with the data, as shown in electronic supplementary material, figure S3. In terms of the categorical response frequencies, the quantal response equilibrium predicts an up-shift for the asymmetric high cost condition and a down-shift for the asymmetric low cost condition for player 1, again in good agreement with the data—compare figure 2*d*.

To investigate further how the equilibrium distributions are reached over time, figure 4*a* shows how the categorical response frequencies change on average over a course of 40 trials. For both players, the initial probability to select either defect or cooperate is 0.5. Over the next 10 trials, this probability gets biased towards the defect action. Crucially, the resulting learning curves for the three different games for player 2 are identical, whereas for player 1 learning is sharper for the asymmetric high cost

experimental data

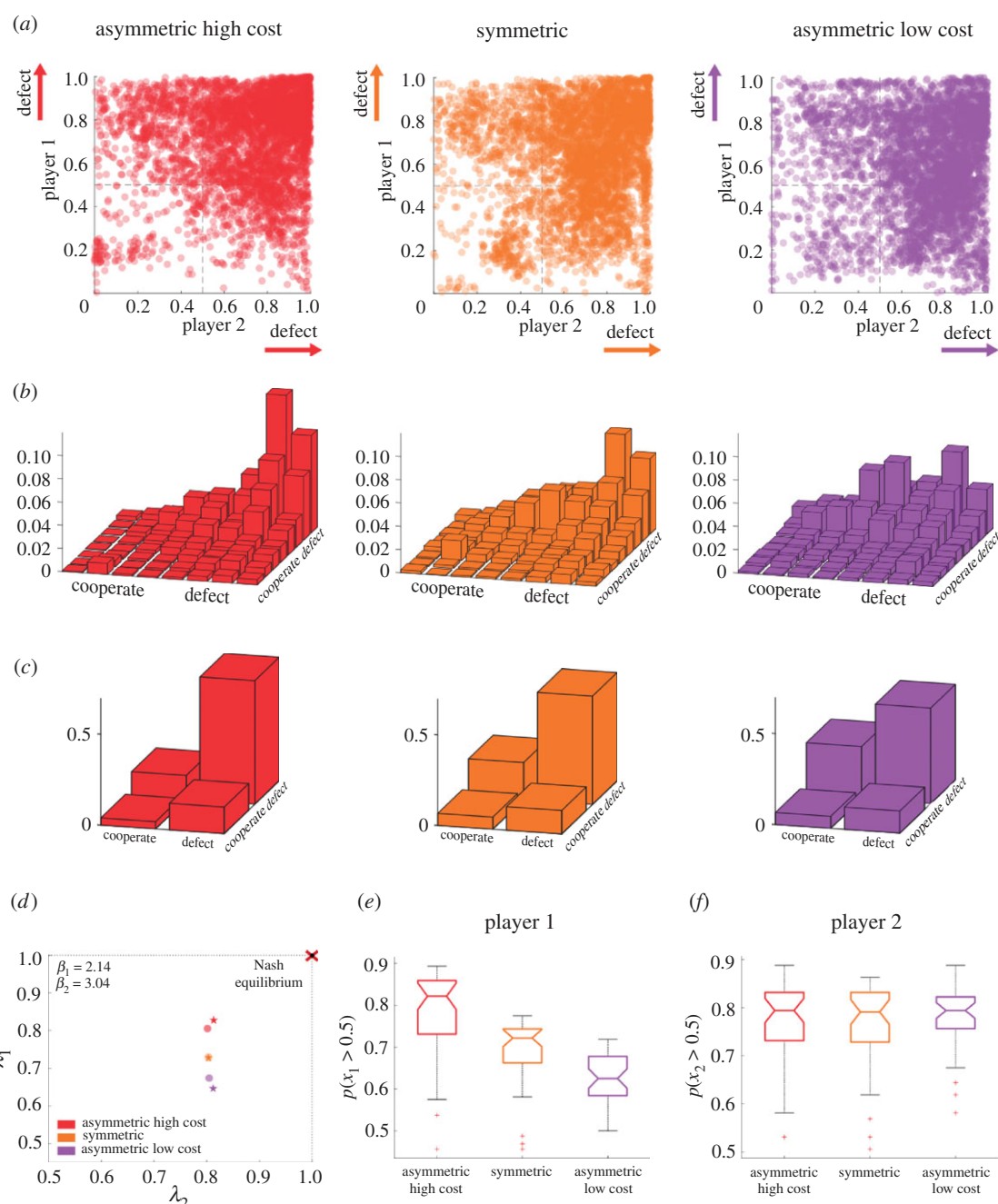

**Figure 2.** Experimental data. (a) Scatter plots of the final decisions of the last 30 trials in the $x_1x_2$-plane for the three Prisoner's Dilemma games, where subjects' actions are expected to cluster around the single pure Nash equilibrium located in the top-right corner at position (1,1). (b) Two-dimensional histograms of the experimental scatter plots. (c) Final decisions with binary categorization. (d) Both players' response frequency $\lambda$ calculated from the categorized actions of the last 30 trials. The experimental data are shown with stars, whereas the circles represent the quantal response equilibria fitted to the data according to equations (6) and (7) with $\beta_1 = 2.14$ and $\beta_2 = 3.04$. (e) Player 1's response frequency of defecting across game conditions. The central marks indicate the median and the edges of the box are the 25th and 75th percentiles considering all trials and subjects. (f) Box plots for player 2's response frequency across conditions. (Online version in colour.)

condition and flatter for the asymmetric low cost condition. Accordingly, player 1 is more indifferent between cooperating and defecting in the asymmetric low cost condition, and more prone to defect in the asymmetric high cost condition, which can also be seen in the temporal evolution of the combined choices—see electronic supplementary material, figure S4. When simulating these learning curves with a pair of continuous reinforcement learning agents employing Q-learning as defined in equation (8) in the electronic supplementary material, this pattern of differentiated learning curves for player 1 across the three games is reproduced—figure 4b.

When analysing the behaviour of the simulated reinforcement learners in the same way as the human players, we can see in the scatter plots the same trend in that the asymmetric high cost condition is more concentrated towards the Nash equilibrium than the asymmetric low cost condition— figure 5a. As for the human subjects, the two-dimensional histograms show a steeper increase in response frequencies for the asymmetric high cost condition and a shallower increase for the asymmetric low cost condition—figure 5b. Finally, when categorizing the responses into (defect–defect), (cooperate–cooperate) and (defect–cooperate or

**Figure 3.** Comparison of the marginal distributions over responses of each player to the equilibrium distributions predicted by the quantal response equilibrium. (*a*) The empirical frequencies of players' responses (solid lines) are compared to the quantal response equilibrium (dash lines) predicting no changes in the response distribution of player 2 and for player 1 elevated response frequencies near the Nash equilibrium in the asymmetric high cost condition, and accordingly suppressed response frequencies around the Nash equilibrium in the asymmetric low cost condition. (*b*) Players' priors represented by histograms over initial positions in the first trials of each block of 40 trials. The solid lines represent simple analytic representations of the priors. (*c*) Quantal response equilibrium considering players' priors (dashed lines) compared to the same data (solid lines). (Online version in colour.)

cooperate–defect), it can be seen that the predominant Nash solution becomes more frequent in the asymmetric high cost condition, and less frequent in the asymmetric low cost condition compared to the classic symmetric game—compare figure 5*c*. The corresponding shifts for the response frequencies of player 1 reproduce the same pattern as observed in the human players. This suggests that reinforcement learning models based on Q-learning cannot only explain convergence to Nash equilibrium solutions [57], but more generally convergence to quantal response equilibria.

## 3. Discussion

In this study, we have investigated the concept of quantal response equilibria in human multi-agent interactions in a continuous sensorimotor version of the symmetric and asymmetric

Prisoner's Dilemma. In particular, we have tested the hypothesis that quantal response equilibria may provide a more accurate description of stable states of human interaction than the prevailing Nash solution concept. During the interactions, subjects were haptically coupled and learned to avoid the haptic coupling force opposing their forward motion, signifying the pay-off for the interaction. In previous studies, it was found that such haptic couplings between two different players in the Prisoner's Dilemma are compatible with the Nash solution, as most interaction endpoints laid in the same quadrant of the workspace than the Nash equilibrium [31]. Similar analyses have also advocated the adequacy of the Nash solution concept for describing sensorimotor interactions in more general scenarios, including mixed equilibrium games like matching pennies [57], coordination games with multiple Nash equilibria like the battle of sexes, chicken or stag hunt [32] as well as Bayesian games that require sensorimotor communication [34].

experimental data

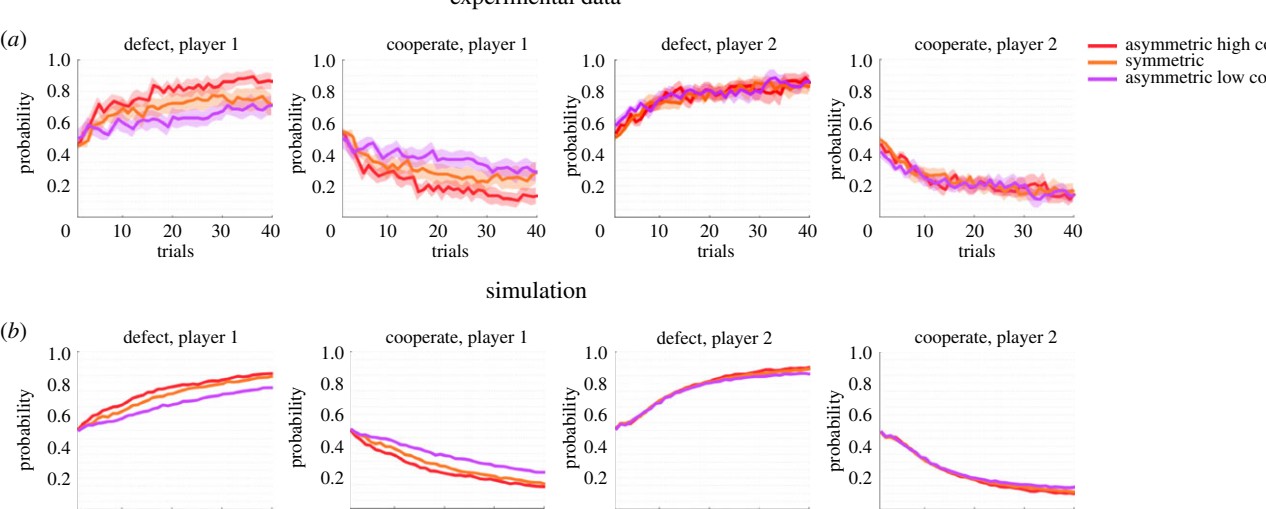

**Figure 4.** Temporal evolution of players' responses. (*a*) Evolution of each player's probability of defecting and cooperating across a set of 40 trials. The shading indicates one standard error of the mean across subjects. (*b*) Evolution of the probability of defecting and cooperating for two simulated Q-learning players.

simulation

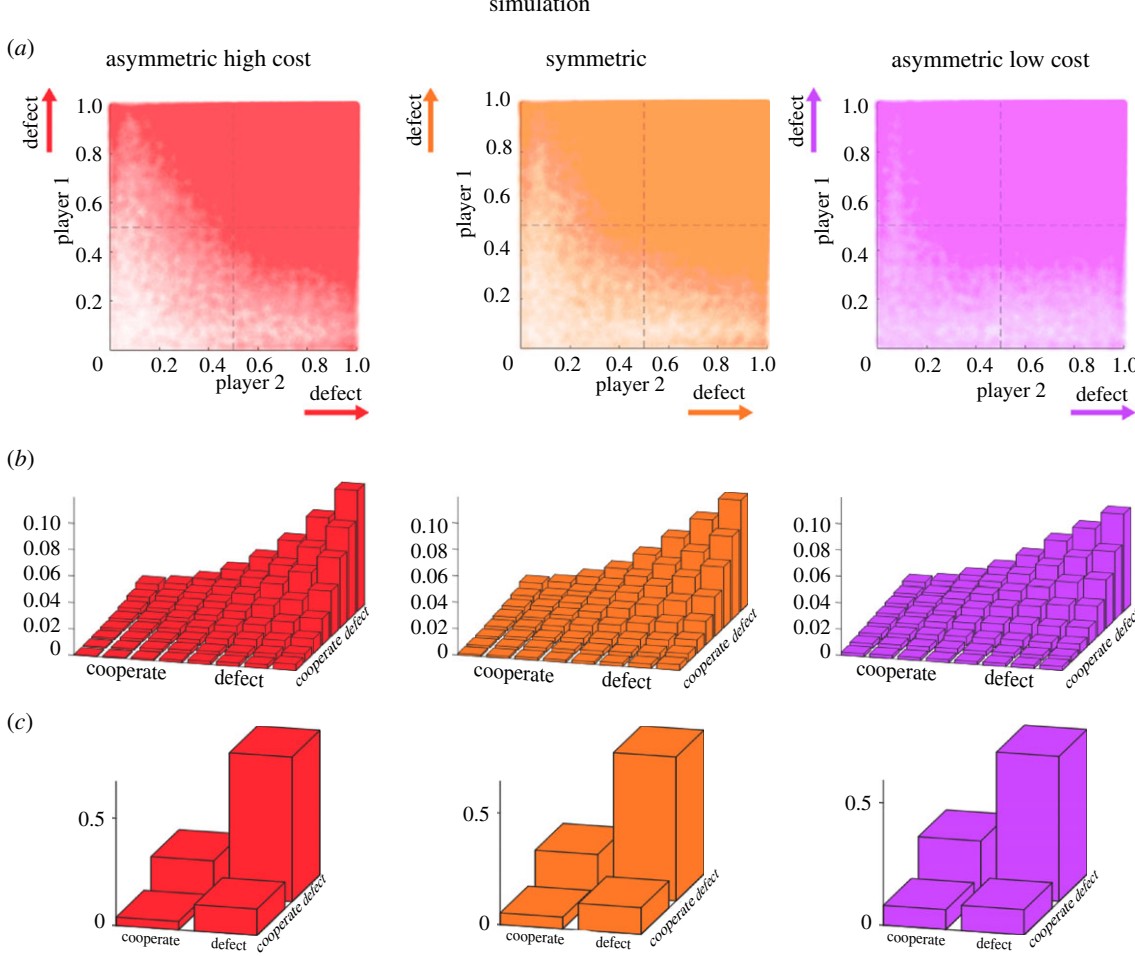

**Figure 5.** Simulated Q-learners. (*a*) Scatter plots of the final decisions in the $x_1 x_2$-plane for the three games, where actions are expected to cluster around the single pure Nash equilibrium located in the top-right corner at position (1,1). (*b*) Two-dimensional histograms of the simulated scatter plots. (*c*) Final decisions in terms of binary categorizations. (Online version in colour.)

Importantly, none of the above studies could distinguish the Nash solution from the quantal response equilibrium, as the two solution concepts are often very close together and perfectly coincide in the absence of computational or precision limits. Accordingly, we have designed a sensorimotor interaction game based on three different 2 × 2-matrix games

corresponding to the classic symmetric form and two asymmetric forms of the Prisoner's Dilemma, thus, allowing for the prediction of a response shift for player 1 in case of the quantal response equilibrium and no such shift in case of the Nash equilibrium. Our results are clearly compatible with the predicted shift and incompatible with the no-shift

prediction of the Nash equilibrium. As the quantal response equilibrium can be seen as a generalization of the Nash equilibrium that contains the Nash solution as a special case in the limit of perfect rationality [58], our results suggest that the quantal response equilibrium should be seen as the more general concept to understand sensorimotor interaction, albeit it will often coincide with the corresponding Nash equilibrium solution.

The most common specification of the quantal response equilibrium model is based on softmax strategies with a single precision parameter [49]. The interpretation of softmax- or logit-strategies in terms of bounded rational choice rules with limited precision in terms of a trade-off between pay-off and entropy has also put the quantal response equilibrium at the heart of bounded rational game theory [58–64]. Extending this trade-off by including prior strategies, bounded rational choice can be described by Boltzmann-like distributions like equations (4) and (5) in the electronic supplementary material, where the individual precision parameter quantifies how much a player is able to deviate from their prior towards a utility maximizing strategy [46,65]. We found in our study that considering players' priors significantly improves the fit predicted by the quantal response equilibrium, especially near the boundaries. Importantly, this is not a result from overfitting by assuming arbitrary priors, but we extracted priors experimentally from the distributions over initial positions at the beginning of each block of trials. This also gives further credence to a growing body of literature that uses utility-information trade-offs to model bounded rational decision-making in the sensorimotor context, emphasizing the role of the prior in such trade-offs [47].

Like the Nash equilibrium, bounded rational response equilibria are defined as fixed points and do not detail any mechanism regarding the decision-making and learning processes that ultimately converge to these fixed points. In our study we have used a continuous Q-learning model with basis functions that was able to reproduce the predicted shifts and the convergence to the quantal response equilibrium. The Q-learner played all three games with the same parameter set. Since the Q-learner is also based on a softmax strategy, it naturally reproduces the predictions of the quantal response equilibrium, because the action probabilities are biased by more or less pay-off in the asymmetric conditions. To study the learning curves we focused on the change of final positions across trials, since we found that initial and final positions within trajectories were generally close. Specifically, we found that in more than 77% of the trials for the symmetric Prisoner's Dilemma, and more than 77% and 78% of the trials for the high and low asymmetric versions, respectively, the players' final decision laid within a 1.6 cm neighbourhood (10% of the workspace) of their initial position in each trial, and there was no systematic change over the block of trials. This suggests that adaptation processes during the trial only had a minor effect and could be neglected.

Our study is part of a broader family of studies that investigate differences between decision-making in sensorimotor tasks and cognitive tasks [66]. The asymmetric Prisoner's Dilemma has been previously investigated in a number of studies where subjects were told the pay-off matrices and they had to make deliberate decisions over the course of a set of repeated games [67–70]. Usually, the aim of the studies was to investigate the effect of the asymmetry on the propensity for cooperation. The results of the studies are difficult to compare due to substantial variations in experimental design, as there are many different ways of introducing asymmetry, for example, affecting all entries or only some entries in the pay-off matrix, or where one player has consistently higher payoffs than the other, or mixed designs, etc. Nevertheless, many of the studies suggest that asymmetry makes reasoning in the game more difficult, and report lower rates of cooperation in asymmetric pay-off conditions [67–70]. In contrast, in our sensorimotor games the frequency of the cooperative solution (cooperate–cooperate) is not modulated by the asymmetry— compare electronic supplementary material, figure S4. Instead, an increase or decrease in the prevalence of the Nash solution (defect–defect) is accompanied by a corresponding decrease or increase in exploitative (cooperate–defect) responses, where player 1 cooperates more or less depending on the asymmetry condition. This is not only in line with the quantal response equilibrium prediction, but also agrees with previous experimental results [31] that have shown that cooperation does not arise as a stable solution during haptic coupling, but only in cognitive versions of the Prisoner's Dilemma involving conscious deliberation. Our current study adds to this previous line of research by highlighting the importance of taking into account limited information processing capabilities due to bounded rationality and how to capture these using information constraints [45–48].

**Ethics.** All experimental procedures were approved by the Ethics Committee of Ulm University and were carried out in accordance with relevant guidelines and regulations.

**Data accessibility.** The data that support the findings of this study are openly available in OPARU at http://dx.doi.org/10.18725/OPARU-38089.

The data are provided in the electronic supplementary material [71].

**Authors' contributions.** C.L.: conceptualization, data curation, formal analysis, investigation, methodology, validation, visualization, writing-original draft, writing-review and editing; G.S.: conceptualization, data curation, formal analysis, investigation, methodology, validation; D.A.B.: conceptualization, formal analysis, funding acquisition, investigation, methodology, project administration, resources, supervision, writing-original draft, writing-review and editing.

All authors gave final approval for publication and agreed to be held accountable for the work performed therein.

**Competing interests.** The authors declare that the research was conducted in the absence of any commercial or financial relationships that could be construed as a potential conflict of interest.

**Funding.** This study was funded by European Research Council (ERC-StG-2015 - ERC Starting Grant, Project ID: 678082, BRISC: Bounded Rationality in Sensorimotor Coordination).

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
