## [Peer Review File · Proceedings of the Royal Society B: Biological Sciences]

Review History

RSPB-2021-1470.R0 (Original submission)

Review form: Reviewer 1

Recommendation

Accept with minor revision (please list in comments)

Scientific importance: Is the manuscript an original and important contribution to its field?

Excellent

General interest: Is the paper of sufficient general interest?

Excellent

Quality of the paper: Is the overall quality of the paper suitable?

Excellent

Is the length of the paper justified?

Yes

Should the paper be seen by a specialist statistical reviewer?

Yes

Do you have any concerns about statistical analyses in this paper? If so, please specify them explicitly in your report.

No

It is a condition of publication that authors make their supporting data, code and materials available - either as supplementary material or hosted in an external repository. Please rate, if applicable, the supporting data on the following criteria.

Is it accessible?

Yes

Is it clear?

Yes

Is it adequate?

Yes

Do you have any ethical concerns with this paper?

No

Comments to the Author

It was a pleasure reading the paper. The study is interesting and the results very useful for readers working in the field of Game Theory under the assumption of bounded rationality of players. Indeed, it provides a physical proof of the inadequacy of rational assumption of decision makers.

I have some minor comments.

1. The way the payoff matrix is presented in Figure 1 of the supplementary information is quite unclear. Indeed, the colors used in the payoff matrix (red and grey) are not the same used for the two players (green and grey), thus making non intuitive the reading. Moreover, the two strategies, cooperate and defect are put in different orders for the two players: defect-cooperate for player 1 and cooperate-defect for player 2. Is there a specific reason for doing that?
2. It is not clear how the position data are collected and recorded. Can the authors explain better this methodology?
3. A more theoretical comment is about the impossibility to have high cooperative strategies, neither pure ones neither mixed ones. It seems that defective strategies (always above 0.5-0.6) compared to cooperative ones, except for case G7 of Figure 3 in the supplementary file. Is a significant level of cooperation of both players never reached?
4. Following the above consideration (point 3), it seems that the best cooperation level is mixed. This is in agreement to the presence of mixed Nash equilibria in some games, such as stag-hunt or chicken games. Did the authors think or plan to perform experiments using the framework of these kind of games?

Review form: Reviewer 2

Recommendation

Reject – article is not of sufficient interest (we will consider a transfer to another journal)

Scientific importance: Is the manuscript an original and important contribution to its field?

Marginal

General interest: Is the paper of sufficient general interest?

Marginal

Quality of the paper: Is the overall quality of the paper suitable?

Marginal

Is the length of the paper justified?

Yes

Should the paper be seen by a specialist statistical reviewer?

No

Do you have any concerns about statistical analyses in this paper? If so, please specify them explicitly in your report.

No

It is a condition of publication that authors make their supporting data, code and materials available - either as supplementary material or hosted in an external repository. Please rate, if applicable, the supporting data on the following criteria.

Is it accessible?

Yes

Is it clear?

N/A

Is it adequate?

N/A

Do you have any ethical concerns with this paper?

No

Comments to the Author

In "Bounded rational response equilibria in human sensorimotor interactions," Lindig-Leon et al. study how humans play a "game" devised by haptically coupling two players. The players must apply a force to hit a target, and the force depends on two real-valued parameters representing the horizontal position at which the force is being applied (one for each player). It is the "game" parameters that determine how the spring constant for each player depends on the current horizontal position parameter of the two players. The authors consider three such parameter matrices, corresponding to a symmetric prisoner's dilemma, an asymmetric prisoner's dilemma with low cost, and an asymmetric prisoner's dilemma with high cost.

The premise for this study is that the Nash equilibrium is the same in all three games, mutual defection. The quantal response equilibrium, taking the form of the popular logit equilibrium, differs across the games and better captures the bounded rationality that humans have been observed to possess in behavioral economics experiments. The authors find that here, too, the players behave in a way that deviates from the Nash prediction and better fits the quantal response prediction.

I am a little torn on what to think of the results. On one hand, I like the experimental design, and the authors note that it is based on a design used in another of their papers. It provides a nice way of obscuring the payoffs and even the game itself from the players, providing them with only abstract feedback based on their actions. (I assume also that the target is chosen randomly to better capture the trajectory, including potential oscillations, that lead to "strategy" collected as the endpoint?) But this is not the main contribution of the present manuscript.

On the other hand, I don't really find the results surprising given how humans have been observed to behave. As the authors note, it is well-known that humans do not always play

according to a Nash equilibrium of a game, and bounded rationality has been influential in explaining deviations from classical economic models. The authors' claim here, assuming I understand it correctly, is that it is unknown whether the findings of behavioral economics in this regard also hold for "sensorimotor" interactions. I am a bit unclear what the major distinction is here. For example, if one were to measure automatic physiological responses in humans, it might be reasonable to assume that a priori these behaviors might differ from those involved in deliberate decision-making. In contrast, the players are instructed to strategically expend as little energy as possible in hitting the target (in 1.5 seconds), which makes the experimental design feel as though its primary purpose is to translate payoff matrices (and the game) into things that cannot be directly observed by the participants. With limited information about the interaction, it then seems natural that traditional descriptions of human actions in behavior economics would be relevant. Is this not the case?

Perhaps the strongest indication to the contrary is in the authors' claim that several studies show that humans who are haptically coupled in this kind of sensorimotor experiment actually do converge to Nash equilibria. Here, however, the referenced articles [7, 8, 19, 25] are authored by the third author of the present study, so initially it is confusing whether the present study is aimed at refuting earlier results or expanding upon them. Later, in the discussion, this is clarified a little, where it is stated that earlier studies tracked only quadrants and could not distinguish between Nash and quantal response equilibria. So, assuming I understand the authors' intentions correctly, this study is not a refutation of commonly held beliefs about sensorimotor interactions, either.

With that said, although I do find the experimental design to be quite clever, my opinion is that the study does not represent a significant enough advance to be of broad interest to the Proceedings B readership.

Decision letter (RSPB-2021-1470.R0)

07-Sep-2021

Dear Dr Lindig:

I am writing to inform you that your manuscript RSPB-2021-1470 entitled "Bounded rational response equilibria in human sensorimotor interactions" has, in its current form, been rejected for publication in Proceedings B.

This action has been taken on the advice of referees, who have recommended that substantial revisions are necessary. With this in mind we would be happy to consider a resubmission, provided the comments of the referees are fully addressed. However please note that this is not a provisional acceptance.

Sincerely,
 Professor Gary Carvalho
 mailto: proceedingsb@royalsociety.org

Associate Editor
 Board Member: 1

Comments to Author:

Reviewer 2 has raised substantive issues regarding the conceptual advance offered by the paper, as well as regarding whether the findings are more appropriately reported in a specialized journal. Given that Reviewer 2 offered praise for the experimental design, we would welcome a resubmission in which revisions address in a substantive and compelling way the concerns raised by Reviewer 2, as well as the points raised by Reviewer 1.

Reviewer(s)' Comments to Author:

Referee: 1

Comments to the Author(s)

It was a pleasure reading the paper. The study is interesting and the results very useful for readers working in the field of Game Theory under the assumption of bounded rationality of players. Indeed, it provides a physical proof of the inadequacy of rational assumption of decision makers.

I have some minor comments.

1. The way the payoff matrix is presented in Figure 1 of the supplementary information is quite unclear. Indeed, the colors used in the payoff matrix (red and grey) are not the same used for the two players (green and grey), thus making non intuitive the reading. Moreover, the two strategies, cooperate and defect are put in different orders for the two players: defect-cooperate for player 1 and cooperate-defect for player 2. Is there a specific reason for doing that?
2. It is not clear how the position data are collected and recorded. Can the authors explain better this methodology?
3. A more theoretical comment is about the impossibility to have high cooperative strategies, neither pure ones neither mixed ones. It seems that defective strategies (always above 0.5-0.6) compared to cooperative ones, except for case G7 of Figure 3 in the supplementary file. Is a significant level of cooperation of both players never reached?
4. Following the above consideration (point 3), it seems that the best cooperation level is mixed. This is in agreement to the presence of mixed Nash equilibria in some games, such as stag-hunt or chicken games. Did the authors think or plan to perform experiments using the framework of these kind of games?

Referee: 2

Comments to the Author(s)

In "Bounded rational response equilibria in human sensorimotor interactions," Lindig-Leon et al. study how humans play a "game" devised by haptically coupling two players. The players must apply a force to hit a target, and the force depends on two real-valued parameters representing

the horizontal position at which the force is being applied (one for each player). It is the “game” parameters that determine how the spring constant for each player depends on the current horizontal position parameter of the two players. The authors consider three such parameter matrices, corresponding to a symmetric prisoner’s dilemma, an asymmetric prisoner’s dilemma with low cost, and an asymmetric prisoner’s dilemma with high cost.

The premise for this study is that the Nash equilibrium is the same in all three games, mutual defection. The quantal response equilibrium, taking the form of the popular logit equilibrium, differs across the games and better captures the bounded rationality that humans have been observed to possess in behavioral economics experiments. The authors find that here, too, the players behave in a way that deviates from the Nash prediction and better fits the quantal response prediction.

I am a little torn on what to think of the results. On one hand, I like the experimental design, and the authors note that it is based on a design used in another of their papers. It provides a nice way of obscuring the payoffs and even the game itself from the players, providing them with only abstract feedback based on their actions. (I assume also that the target is chosen randomly to better capture the trajectory, including potential oscillations, that lead to “strategy” collected as the endpoint?) But this is not the main contribution of the present manuscript.

On the other hand, I don’t really find the results surprising given how humans have been observed to behave. As the authors note, it is well-known that humans do not always play according to a Nash equilibrium of a game, and bounded rationality has been influential in explaining deviations from classical economic models. The authors’ claim here, assuming I understand it correctly, is that it is unknown whether the findings of behavioral economics in this regard also hold for “sensorimotor” interactions. I am a bit unclear what the major distinction is here. For example, if one were to measure automatic physiological responses in humans, it might be reasonable to assume that a priori these behaviors might differ from those involved in deliberate decision-making. In contrast, the players are instructed to strategically expend as little energy as possible in hitting the target (in 1.5 seconds), which makes the experimental design feel as though its primary purpose is to translate payoff matrices (and the game) into things that cannot be directly observed by the participants. With limited information about the interaction, it then seems natural that traditional descriptions of human actions in behavior economics would be relevant. Is this not the case?

Perhaps the strongest indication to the contrary is in the authors’ claim that several studies show that humans who are haptically coupled in this kind of sensorimotor experiment actually do converge to Nash equilibria. Here, however, the referenced articles [7, 8, 19, 25] are authored by the third author of the present study, so initially it is confusing whether the present study is aimed at refuting earlier results or expanding upon them. Later, in the discussion, this is clarified a little, where it is stated that earlier studies tracked only quadrants and could not distinguish between Nash and quantal response equilibria. So, assuming I understand the authors’ intentions correctly, this study is not a refutation of commonly held beliefs about sensorimotor interactions, either.

With that said, although I do find the experimental design to be quite clever, my opinion is that the study does not represent a significant enough advance to be of broad interest to the Proceedings B readership.

Author's Response to Decision Letter for (RSPB-2021-1470.R0)

See Appendix A.

RSPB-2021-2094.R0

Review form: Reviewer 2

Recommendation

Accept with minor revision (please list in comments)

Scientific importance: Is the manuscript an original and important contribution to its field?

Good

General interest: Is the paper of sufficient general interest?

Good

Quality of the paper: Is the overall quality of the paper suitable?

Good

Is the length of the paper justified?

Yes

Should the paper be seen by a specialist statistical reviewer?

No

Do you have any concerns about statistical analyses in this paper? If so, please specify them explicitly in your report.

No

It is a condition of publication that authors make their supporting data, code and materials available - either as supplementary material or hosted in an external repository. Please rate, if applicable, the supporting data on the following criteria.

Is it accessible?

N/A

Is it clear?

N/A

Is it adequate?

N/A

Do you have any ethical concerns with this paper?

No

Comments to the Author

The authors have done a good job in concisely highlighting the main contribution of this paper in relation to their previous works, which I thought was not clear in the original submission. The distinction between this experimental setup and studies of bounded rationality in behavioral economics was also helpful, although I am not entirely convinced that the underlying mechanisms are that different. With that being said, I am happy to support publication of the revision in Proceedings B, as it does add a valuable contribution to studies of human behavior in social dilemmas.

One thing that I would ask is that the statistical details reported in the text (e.g. figure 1) be clearly explained in the supplement. This should require only a minor addition to the SI.

Decision letter (RSPB-2021-2094.R0)

06-Oct-2021

Dear Dr Lindig

I am pleased to inform you that your manuscript RSPB-2021-2094 entitled "Bounded rational response equilibria in human sensorimotor interactions" has been accepted for publication in Proceedings B.

The referee(s) have recommended publication, but also suggest some minor revisions to your manuscript. Therefore, I invite you to respond to the referee(s)' comments and revise your manuscript. Because the schedule for publication is very tight, it is a condition of publication that you submit the revised version of your manuscript within 7 days. If you do not think you will be able to meet this date please let us know.

Sincerely,
Professor Gary Carvalho
mailto: proceedingsb@royalsociety.org

Associate Editor
Board Member
Comments to Author:

We would like to thank the authors for nicely addressing prior reviewer concerns. One minor revision has still been suggested. We congratulate the authors for their distinctive contribution.

Reviewer(s)' Comments to Author:

Referee: 2

Comments to the Author(s).

The authors have done a good job in concisely highlighting the main contribution of this paper in relation to their previous works, which I thought was not clear in the original submission. The distinction between this experimental setup and studies of bounded rationality in behavioral economics was also helpful, although I am not entirely convinced that the underlying mechanisms are that different. With that being said, I am happy to support publication of the revision in Proceedings B, as it does add a valuable contribution to studies of human behavior in social dilemmas.

One thing that I would ask is that the statistical details reported in the text (e.g. figure 1) be clearly explained in the supplement. This should require only a minor addition to the SI.

Author's Response to Decision Letter for (RSPB-2021-2094.R0)

See Appendix B.

Decision letter (RSPB-2021-2094.R1)

13-Oct-2021

Dear Dr Lindig

I am pleased to inform you that your manuscript entitled "Bounded rational response equilibria in human sensorimotor interactions" has been accepted for publication in Proceedings B.

Your article has been estimated as being 9 pages long. Our Production Office will be able to confirm the exact length at proof stage.

Data Accessibility section

Open Access

Paper charges

You are allowed to post any version of your manuscript on a personal website, repository or preprint server. However, the work remains under media embargo and you should not discuss it

with the press until the date of publication. Please visit <https://royalsociety.org/journals/ethics-policies/media-embargo> for more information.

Sincerely,
Editor, Proceedings B
mailto: proceedingsb@royalsociety.org

Appendix A

We thank the reviewers for their comments that have helped us to clarify the manuscript. We have made amendments to the manuscript in line with the reviewers' comments. In particular, we have remodelled the introduction with an improved motivation that clarifies the contribution of the paper within the wider literature.

Referee: 1

Comments to the Author(s)

It was a pleasure reading the paper. The study is interesting and the results very useful for readers working in the field of Game Theory under the assumption of bounded rationality of players. Indeed, it provides a physical proof of the inadequacy of rational assumption of decision makers.

I have some minor comments.

1. The way the payoff matrix is presented in Figure 1 of the supplementary information is quite unclear. Indeed, the colors used in the payoff matrix (red and grey) are not the same used for the two players (green and grey), thus making non intuitive the reading. Moreover, the two strategies, cooperate and defect are put in different orders for the two players: defect-cooperate for player 1 and cooperate-defect for player 2. Is there a specific reason for doing that?

We thank the reviewer for pointing out the inconsistencies in the display, which we have mended in the revised version. Please note that on a numerical scale, the strategies for the two players are in the same order (for example, 0/0 corresponds to cooperate/cooperate and 1/1 corresponds to defect/defect), which means that in the matrix cooperate/cooperate has to be in the bottom left (which makes it appear in the “wrong” order when reading from top to bottom).

2. It is not clear how the position data are collected and recorded. Can the authors explain better this methodology?

We have added more details about the recording methodology in the methods section of the supplementary material to clarify how endpoints were determined from the recorded trajectories.

3. A more theoretical comment is about the impossibility to have high cooperative strategies, neither pure ones neither mixed ones. It seems that defective strategies (always above 0.5-0.6) compared to cooperative ones, except for case G7 of Figure 3 in the supplementary file. Is a significant level of cooperation of both players never reached?

In all the groups we have tested so far, the cooperate/cooperate solution has never been stable, at most we have observed subject pairs like G7, where one player is very close to random and the other player defects.

4. Following the above consideration (point 3), it seems that the best cooperation level is mixed. This is in agreement to the presence of mixed Nash equilibria in some games, such as stag-hunt or chicken games. Did the authors think or plan to perform experiments using the framework of these kind of games?

In previous experiments (Braun et al. Motor coordination: when two have to act as one, Exp. Brain Res. 2011, and with human vs. artificial players in Grau-Moya et al. The effect of model uncertainty on cooperation in sensorimotor interactions, Proc Roy Soc Int, 2013) we have tested the effect of mixed equilibria and multiple equilibria on cooperation, but please note that none of these previous designs were suitable to distinguish Nash equilibrium solutions from QRE solutions.

We thank the reviewers for their comments that have helped us to clarify the manuscript. We have made amendments to the manuscript in line with the reviewers' comments. In particular, we have remodelled the introduction with an improved motivation that clarifies the contribution of the paper within the wider literature.

Referee: 2

Comments to the Author(s)

In "Bounded rational response equilibria in human sensorimotor interactions," Lindig-Leon et al. study how humans play a "game" devised by haptically coupling two players. The players must apply a force to hit a target, and the force depends on two real-valued parameters representing the horizontal position at which the force is being applied (one for each player). It is the "game" parameters that determine how the spring constant for each player depends on the current horizontal position parameter of the two players. The authors consider three such parameter matrices, corresponding to a symmetric prisoner's dilemma, an asymmetric prisoner's dilemma with low cost, and an asymmetric prisoner's dilemma with high cost.

The premise for this study is that the Nash equilibrium is the same in all three games, mutual defection. The quantal response equilibrium, taking the form of the popular logit equilibrium, differs across the games and better captures the bounded rationality that humans have been observed to possess in behavioral economics experiments. The authors find that here, too, the players behave in a way that deviates from the Nash prediction and better fits the quantal response prediction.

I am a little torn on what to think of the results. On one hand, I like the experimental design, and the authors note that it is based on a design used in another of their papers. It provides a nice way of obscuring the payoffs and even the game itself from the players, providing them with only abstract feedback based on their actions. (I assume also that the target is chosen randomly to better capture the trajectory, including potential oscillations, that lead to "strategy" collected as the endpoint?) But this is not the main contribution of the present manuscript.

Yes, the target position is indeed randomized from trial to trial to keep subjects attention for every decision. In terms of experimental design, the main novelty of the paper is the design of the two asymmetric prisoner dilemma games and their translation into sensorimotor games, as only this design allows for the discrimination between the Nash and the QRE prediction. The previous designs did not allow for this distinction. The main contribution of the paper is to investigate subjects' behaviour in these two games and compare to the symmetric prisoners' dilemma to test the prediction (Nash vs QRE).

On the other hand, I don't really find the results surprising given how humans have been observed to behave. As the authors note, it is well-known that humans do not always play according to a Nash equilibrium of a game, and bounded rationality has been influential in explaining deviations from classical economic models. The authors' claim here, assuming I understand it correctly, is that it is unknown whether the findings of behavioral economics in this regard also hold for "sensorimotor" interactions. I am a bit unclear what the major distinction is here. For example, if one were to measure automatic physiological responses in humans, it might be reasonable to assume that a priori these behaviors might differ from those involved in deliberate decision-making. In contrast, the players are instructed to strategically expend as little energy as possible in hitting the target (in 1.5 seconds), which makes the experimental design feel as though its primary purpose is to translate payoff matrices (and the game) into things that cannot be directly observed by the participants. With limited information about the interaction, it then seems natural that traditional descriptions of human actions in behavior economics would be relevant. Is this not the case?

We thank the reviewer for raising the question about sensorimotor decision-making and its relation to other studies in behavioural economics. Accordingly, we have amended the Introduction to clarify this relationship. Our study is part of a broader family of studies that have investigated differences between decision-making in sensorimotor tasks and cognitive tasks---see for example the review chapter A. Toga S.-W. Wu, M. R. Delgado, L. T. Maloney, "Motor decision-making" in Brain Mapping: An Encyclopedic Reference, A. Toga, Ed. (Elsevier Science & Technology, 2015), pp. 417–427. While studies in behavioural economics often focus on cognitive tasks in decision problems with explicitly communicated utilities (often in terms of monetary payoffs) and clearly defined and known uncertainties (often stated explicitly as probabilities), sensorimotor tasks typically involve implicit, action-related utilities (often in terms of motor effort or task accuracy) and experiential probabilities that have to be learnt from many repetitions. Moreover, motor tasks often involve implicit learning (e.g. how to ride a bike) in contrast to explicit learning (e.g. involving cognitive strategies when learning how to play the boardgame monopoly). In our setup the characteristic feature is that the two players influence each other's behaviour continuously in time through force coupling with continuous action spaces over repeated trials (experiential uncertainty). In contrast, two-player interactions considered in classical game theory are typically thought to involve cognition in games with discrete actions and discrete time steps for decision-making such as tic-tac-toe, the ultimatum game or the prisoner's dilemma---see Braun et al. 2009, PLoS Comp Biol. To underline the existence of some interesting differences between the two kinds of games (sensorimotor vs. cognitive), we found, for example, that sensorimotor interactions regularly converged to the predicted Nash solution of (defect,defect), whereas cognitive versions of the prisoners' dilemma regularly lead to some level of cooperation.

Other studies have also found interesting differences between economic decision tasks and their equivalent sensorimotor tasks that have been communicated to broad audiences (e.g. Wu, Delgado, Maloney, Economic decision-making compared with an equivalent motor task, PNAS, 2009). In particular, it has often been found that human

sensorimotor behaviour abides by rational decision-making models (see for example the review in Kording, 2007, Science, What should the nervous system do?), whereas for economic studies deviations from rational behaviour have been more routinely reported---although this idea has also been contested, see for example Jarvstad, Hahn, Rushton, Warren, Perceptuo-motor, cognitive, and description-based decision-making seem equally good, PNAS, 2013. From the viewpoint of this debate, it is an interesting question whether quantal response equilibria that have been found to capture behaviour in economic decision tasks are also applicable to sensorimotor tasks, or whether rational concepts like the Nash equilibrium are adequate to capture sensorimotor interactions more generally (as has been suggested for maximum expected utility models in single agent sensorimotor tasks). Finally, our results do not simply replicate QRE findings from behavioural economics in a sensorimotor task, but we found that the quantal response equilibria are sensitive to strategy priors (see Figure 2), which is a refinement of the original QRE concept and therefore provides an interesting result in its own right.

Perhaps the strongest indication to the contrary is in the authors' claim that several studies show that humans who are haptically coupled in this kind of sensorimotor experiment actually do converge to Nash equilibria. Here, however, the referenced articles [7, 8, 19, 25] are authored by the third author of the present study, so initially it is confusing whether the present study is aimed at refuting earlier results or expanding upon them. Later, in the discussion, this is clarified a little, where it is stated that earlier studies tracked only quadrants and could not distinguish between Nash and quantal response equilibria. So, assuming I understand the authors' intentions correctly, this study is not a refutation of commonly held beliefs about sensorimotor interactions, either.

With that said, although I do find the experimental design to be quite clever, my opinion is that the study does not represent a significant enough advance to be of broad interest to the Proceedings B readership.

In previous studies we have indeed found that the Nash equilibrium concept was adequate to describe subjects' behaviour. Importantly, however, this was not a consequence of an insufficient analysis of the data in the previous studies, but a consequence of the experimental design, as in all games we have tested so far the Nash equilibrium and the QRE coincide and make the same predictions. Also, the Nash concept for sensorimotor interactions has found broadly interest and has been prominently reviewed (see for example, Wolpert, Diedrichsen, Flanagan, Principles of sensorimotor learning, Nat Rev Neurosci, 2011), but has not considered the possibility of the QRE. The main novelty of our current study is therefore an experimental design that can distinguish between the two. In that sense, one could argue that we "refute" the previous stipulation that Nash equilibria may provide a general tool to capture sensorimotor interactions, even though our previous result that the Nash equilibrium captures sensorimotor behaviour in the particular games that we tested is perfectly fine (since it is the same as the QRE solution in these games).

Appendix B

We thank the reviewers for their responses and have adapted the results to include the details of the statistical evaluation.